# Changes in U.S. medical school conflict of interest policies from 2014 to 2023

Shamik Bhat[1]*, Devika A. Shenoy[2], Magda Wojtara[3], Alissa Kainrath[4], Oak Sonfist[5], Jantzen Faulkner[6], Brianna Wang[7], Linnea Wilson[8], Timothy S. Anderson[9]

**1** School of Medicine, Yale University, New Haven, Connecticut, United States of America, **2** School of Medicine, Duke University, Durham, North Carolina, United States of America, **3** David Geffen School of Medicine, University of California, Los Angeles, California, United States of America, **4** Chicago Medical School, Rosalind Franklin University, North Chicago, Illinois, United States of America, **5** Pacific Northwest University of Health Sciences, Yakima, Washington, United States of America, **6** Center for Health Sciences, Oklahoma State University, Tulsa, Oklahoma, United States of America, **7** Division of General Medicine, Beth Israel Deaconess Medical Center, Boston, Massachusetts, United States of America, **8** Division of General Medicine, Beth Israel Deaconess Medical Center, Boston, Massachusetts, United States of America, **9** Division of General Internal Medicine, University of Pittsburgh, Pittsburgh, Pennsylvania, United States of America

* shamik.bhat@yale.edu

## Abstract

### Background

Concerns about the influence of the pharmaceutical industry on medical education, ranging from education of students to professional development, have led professional societies to recommend regulation of interactions between industry and medical schools. The objective of this study was to evaluate conflict of interest (COI) policies at medical schools in 2023 compared to 2014.

### Methods

This study used a cross-sectional design to evaluate the COI policies at the top 30 medical schools identified by US News and World Report rankings. The authors collected policies by survey and review of public websites, and assessed their quality across 15 domains informed by guidelines published by leading national organizations and previous PharmFree Scorecards. Each domain was graded on a 3-level scale derived from professional organization guidelines, which when totaled corresponded to the following letter grades: an "A" (score 38–45), "B" (32–37), "C" (25–31), and "I/F" (< 25). This study assessed industry payments to school leadership using the 2023 Open Payments database.

### Results

Eleven of thirty medical schools submitted COI policies, and the remainder were analyzed based on publicly available information. No school received an "A," 22 (73.3%)

**Data availability statement:** All relevant data are within the manuscript and its Supporting Information files.

**Funding:** This project, conducted through the American Medical Student Association, was funded by Arnold Ventures Grant ID 22-08026.

**Competing interests:** The authors have declared that no competing interests exist.

schools received a "B", 6 (20.0%) schools received a "C", and 2 (6.7%) schools received an "I/F". Most schools had model policies around COI enforcement (29/30, 96.7%), gift acceptance (25/30, 83.3%), and ghostwriting (24/30, 80.0%). No schools had model policies in limiting direct faculty payments. When comparing 2014 and 2023 Scorecards over the shared 14 domains, 14 (46.7%) schools had a decrease in score, 11 (36.7%) schools had an increase, and 5 (16.7%) schools had no change. Faculty at every school accepted industry payments, including 20 (16.7%) deans and 52 (19.3%) of clerkship directors.

## Conclusions

Medical school COI policies remain less stringent than consensus recommendations; thus, renewed attention to policies and implementation is needed to ensure bias-free medical education.

## Introduction

With the rising costs of both medical education and healthcare—and increased public-private partnerships in the United States in the wake of the COVID-19 pandemic—potential relationships between the pharmaceutical industry and American medical institutions deserve closer evaluation.[1,2] Industry payments to medical providers can lead to unintended consequences, ranging from individual bias to shifting healthcare decisions.[3–5] For example, pharmaceutical industry-sponsored meals have been associated with increased physician prescribing rates of promoted medications in comparison to alternatives in their same drug class.[4] Additionally, analyzing Medicare Part D data revealed that increased opioid-related payments were correlated with increased prescribed daily doses of oxycodone, hydrocodone, and fentanyl.[3]

Financial relationships between industry and clinicians manifest early and consistently in medical training: [6] researchers have noted correlations between resident interaction with pharmaceutical representatives and changes in prescribing habits.[7] Continuing medical education (CME), which allows physicians to further develop professional skills in practice, sees similar trends.[8] Around a quarter of CME funding is industry-provided, leading to predictable consequences: physicians' prescribing habits are changed, formulary changes are more likely, adverse effects are de-emphasized, and patients' trust in physicians may become threatened.[9–14] These ties to the pharmaceutical industry may play a role in influencing medical practices, prompting a closer look at their effects on medical education, prescribing practices, and American healthcare as a whole.

National organizations such as the American Association of Medical Colleges (AAMC) and the National Academy of Medicine (NAM) have published established, evidence-based standards that are designed to limit industry influence in medical education with thorough conflict of interest (COI) policies.[15,16] The American Medical Student Association (AMSA) uses these standards to develop a grading methodology for medical school COI policies and has periodically released evaluations of medical

school COI policies, referred to as *AMSA PharmFree Scorecards.* Much has changed in medicine since 2014 when the last AMSA Scorecard was published: the COVID-19 pandemic saw increased cooperation between academic institutions and pharmaceutical firms to develop vaccines and therapeutics, potentially increasing students' exposure to industry. Firms have also relied on academic institutions to insulate them from financial risk in research, and virtual interactions between industry and physicians have increased.[17] Thus, this study sought to evaluate the COI policies of 30 prestigious medical institutions using similar methodology as previous Scorecard iterations.[18] Additionally, as the Centers for Medicare & Medicaid Services (CMS) Open Payments program now allows for examination of financial relationships between industry and individual physician, an analysis of industry payments to medical school leadership was performed.[19] This analysis also included deans and clerkship directors, who often hold significant influence in developing clinical curricula.

## Methods

### Overview

This study used a cross-sectional analysis of COI policies of 30 medical institutions, chosen as the top ranked by the 2022 iteration of US News and World Report "Best Medical Schools: Research" list.[18] The report's rankings are based on assessments by medical school deans and residency program directors, student scores, and federal research grants received.[20] We utilized the US News and World Report rankings to ensure consistency in the methodology between the prior and current iterations of the scorecard. Evaluation of COI policies included 14 domains from the 2014 AMSA Scorecard and one new domain: direct industry payments to faculty documented on Open Payments.[19] We then compared school policies in 2023 to those from 2014, overall and by individual domain. Similar to prior iterations of the AMSA Scorecard, this study did not require formal ethics approval due to the lack of human subjects, material, or data.[21]

### Data collection

A standardized email and survey (Supporting Information S1 Appendix) were sent to the Dean of Curriculum (or equivalent role) at each of the top 30 medical schools (Supporting Information S2 Table). Surveys and reminders were sent throughout July and August 2023, and for schools that did not complete the survey, manual data collection using publicly available policies took place. Schools were informed of our intent to publish an analysis of their submitted policies, and informed that if no response was received, we would conduct analysis on publicly available policies (Supporting Information S1 Appendix). Supporting Information S2 Table contains citations for all analyzed policies. Data from the 2014 iteration of the AMSA Scorecard was obtained through AMSA.

### Grading criteria

The most recent prior iteration of the AMSA Scorecard analyzed COI policies enacted in or before 2014.[21] The 2023 revision uses a similar grading and scoring approach, which uses consensus recommendations published by the AAMC and other professional organizations (Supporting Information S2 Appendix).[16,21] Scoring criteria for the original 14 domains remained unchanged between the 2014 and 2023 iterations, with one exception. In the 2014 iteration for the ghostwriting domain, "1 point" was assigned for no policy discouraging or prohibiting ghostwriting, "2 points" for discouragement of ghostwriting, and "3 points" for prohibition of ghostwriting. Ghostwriting, which allows industry representatives to author or influence scholarly articles and educational materials, can hide conflicts of interest and bias medical guidelines.[22] To better emphasize the meaningful difference between prohibition and discouragement of ghostwriting, the grading was simplified to, "1 point" is assigned for lack of prohibition, including discouragement, and "3 points" for prohibition of ghostwriting (Supporting Information S3 Appendix).

The 2023 iteration of the Scorecard added one new category of industry influence in its 15th domain: direct payments from industry to leadership at medical schools, identified using Open Payments. These details on financial ties were

unavailable in 2014 and allow further analysis of industry connections to medical schools. [10] This study analyzed payments to 13 faculty members at each school that often frequently interact with students and have significant influence on curricula: deans of the medical schools, deans of education, deans of diversity, deans of research, as well as clerkship directors for family medicine, internal medicine, pediatrics, neurology, psychiatry, surgery, OB/GYN, radiology, and emergency medicine. We identified faculty members on school websites using the full name and state of the institution in question and their financial ties were analyzed regardless of teaching status.

Two trained analysts independently graded policies relative to AMSA model policies. For each domain, a score of "3" was assigned for model policies, a "2" for moderate policies that are making progress toward the model, a "1" for inadequate policies that do not address the domain's primary concern, and a "0" for no relevant policy. We calculated payments to medical school leadership using data from the 2023 Open Payments database, which reports payments received by physicians in the 2023 calendar year. Leadership was scored according to the proportion of faculty at every school with $0 in industry payments (3 points), above $0 and less than $5,000 (2 points), and greater than/equal to $5,000 (1 point). Federal rules published by the Department of Health and Human Services generally define yearly contributions of $5,000 as reaching the threshold of "significant financial interests" for researchers and clinicians. We adopt the $5,000 yearly benchmark to remain consistent with this guidance.[23,24] Trained research assistants (BW, LW) independently assessed and graded each policy and provided a justification for their scoring; the team then confirmed each score based on the grading methodology and the analyst's justification. An overall score was calculated by adding up scores in each domain, with a maximum potential score of 45. This score was used to assign a letter grade for each school. An "A" was assigned for schools receiving 38 points or higher, a "B" for schools receiving a score of 32–37, and a "C" for schools receiving a score of 25–31. For schools receiving lower than 25 points, an "I/F" (incomplete/failed) was assigned. These conversions of raw score (up to 45 points) to letter grades were consistent with the 2014 iteration, ensuring consistent comparisons can be made across iterations.

## IRB statement

Given the lack of human subjects' data, this study did not undergo review with an Institutional Review Board (IRB).

## Statistical analysis

Descriptive statistics were used to characterize overall scores and the proportion of schools with model, moderate, inadequate and no policy for each domain. A paired two-sample $t$-test compared total school scores and average domain scores between the 2014 and 2023 Scorecards. An unpaired $t$-test compared the scores among schools that submitted policies directly to the PharmFree team and those that did not. Separate calculations were made excluding domain 15 (industry funding to leadership), as this was not assessed in 2014. A sensitivity analysis which excluded domain 7 (ghostwriting) was conducted as the scoring criteria for this domain changed slightly between iterations. The median and interquartile range (IQR) for leadership receiving industry payments were calculated; these results were separated by the three types of leadership included (dean of medical school, other deans, and clerkship directors).

## Results

Eleven of the 30 medical schools provided updated COI policies in 2023 (response rate 36.7%) and the remainder of policies were identified based on publicly available policies. In the 2023 iteration, no school received an "A" grade, 22 (73.3%) schools received a "B" grade, 6 (20.0%) schools received a "C" grade, and 2 (6.7%) schools received an "I/F" (Fig 1). There were no significant differences in scores between schools that submitted materials and schools that did not (mean score 34.3 versus 32.2, $p = 0.11$) (Supporting Information S3 Table).

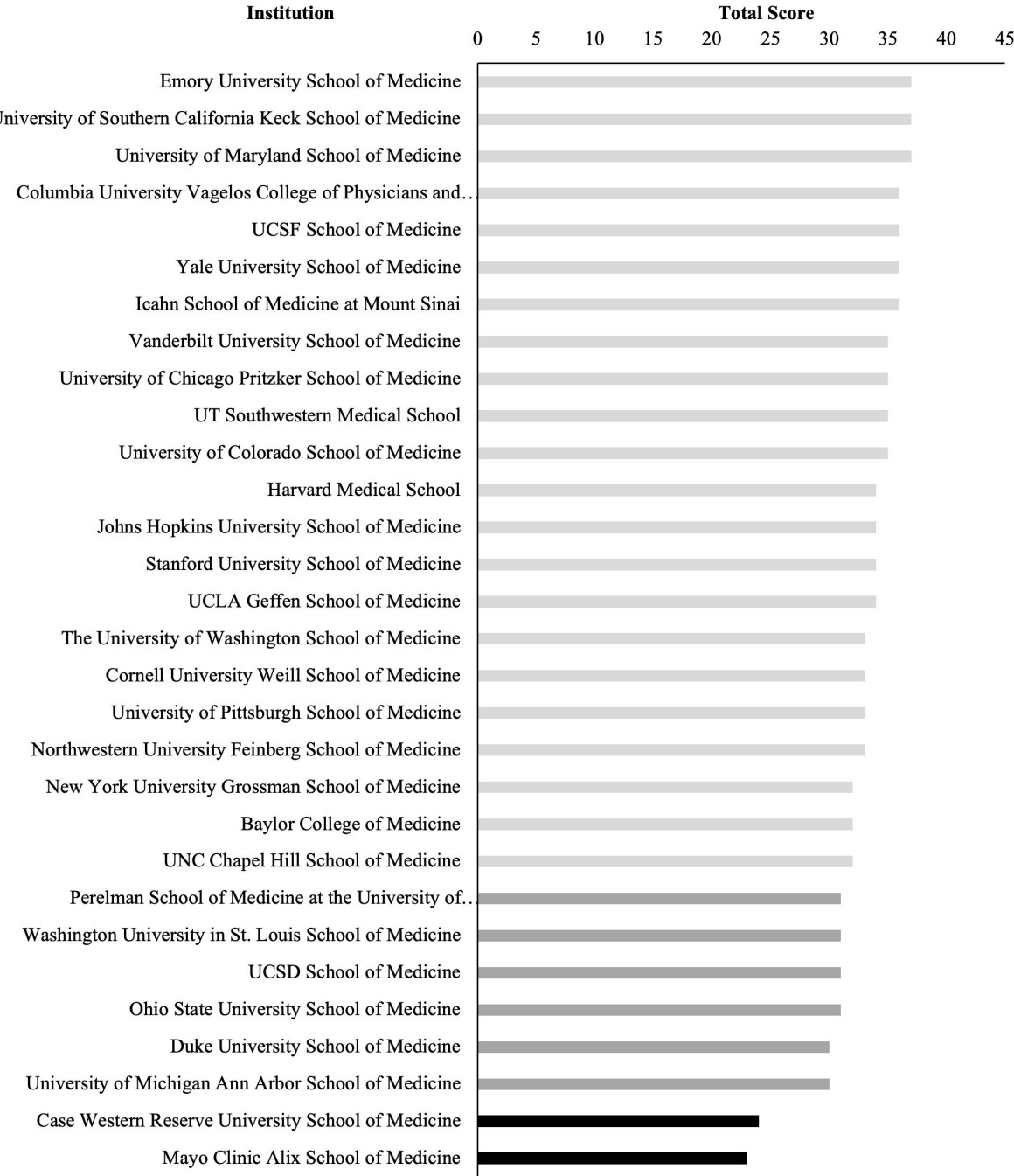

**Fig 1. Numerical Scores for all Evaluated Schools, 2023.** Total scores were calculated for all included institutions. The maximum possible score is 45. We scored each of the 15 domains from 0-3, where 0 was assigned for missing policies, 1 for inadequate policies, 2 for moderate policies, and 3 for model policies. Schools colored in light grey received a "B", schools colored in dark grey received a "C", and schools colored in black received a "I/F".

## Distribution of model policies

Most schools had model policies related to enforcement of COI policies (29/30, 96.7%), prohibiting acceptance of gifts (25/30, 83.3%), ghostwriting (24/30, 80.0%), and promotional speaking relationships (21/30, 70%) (**Fig 2**). Model policies on presence of industry representatives (17/30, 56.7%), and COI disclosure (17/30, 56.7%) were enacted for

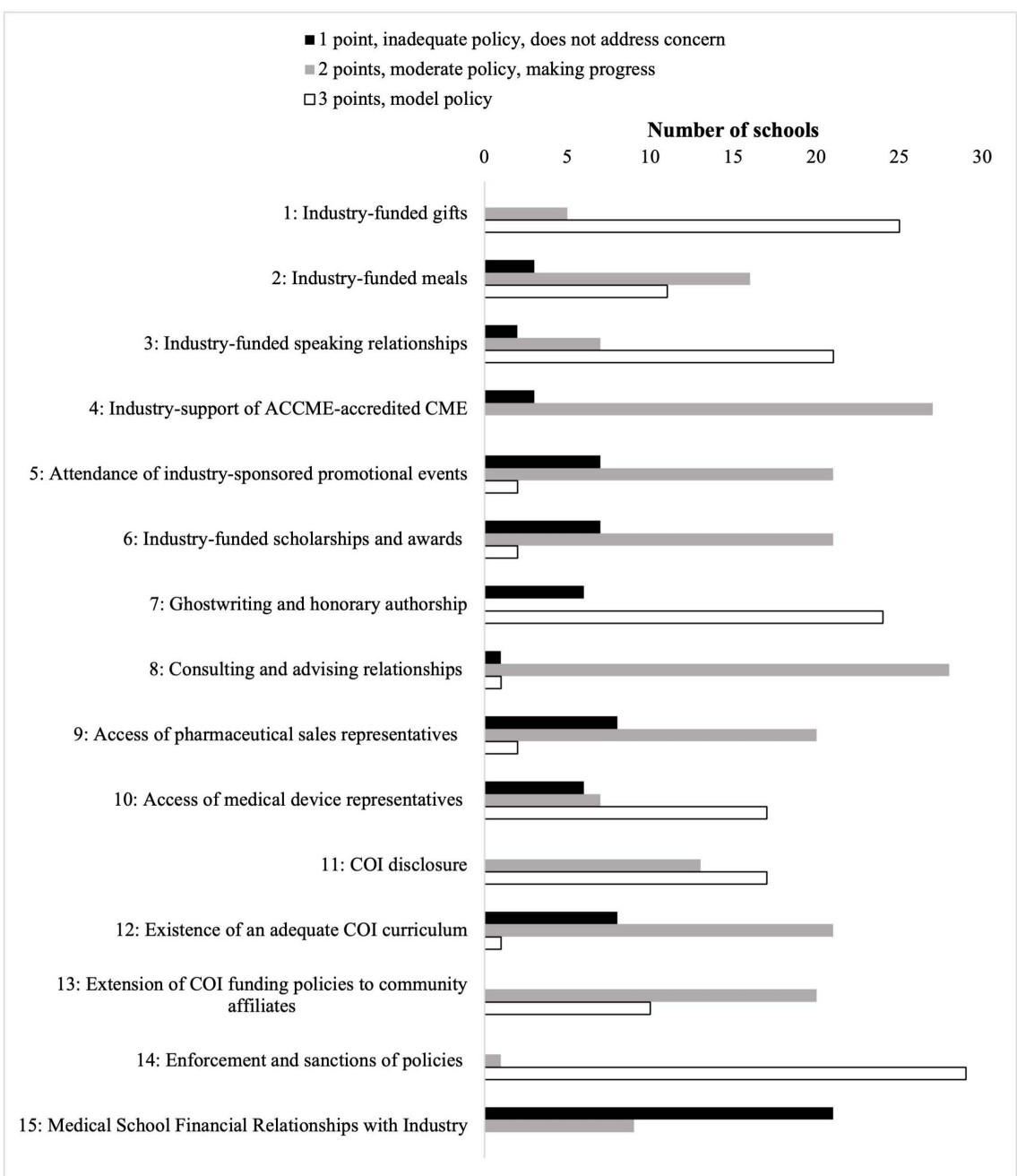

**Fig 2. AMSA Scores Assigned by Domain, 2023.** The number of schools that have inadequate policies that do not address the domain (black, scored 1 point), moderate policies that are making progress towards model policies (gray, scored 2 points), or model (white, black outline) policies within each of the evaluated domains.

approximately half of schools. In contrast, no schools had model policies related to industry payments to faculty (0/30, 0.0%) or industry support of continuing medical education (0/30, 0.0%). One school had model policies around faculty having consulting relationships (1/30, 3.3%), and two had model policies around promotional events (2/30, 6.7%) and presence of pharmaceutical sales representatives on campus (2/30, 6.7%).

### Industry payments to medical school leadership

As shown in **Fig 2**, every school had one or more medical school leaders who received industry payments. Payments varied across positions, with medical school deans receiving a median of $165,002.22 in 2023. (IQR $272,193.28) (**Fig 3**); 21 schools (70.0%) had combined payments to faculty in 2023 greater than $5,000. Of the 120 deans analyzed (four at each institution), 20 (16.7%) accepted industry funding, 14 (11.6%) of whom accepted more than $5,000. Of the 270 clerkship directors analyzed (nine at each institution), 52 (19.3%) accepted industry funding in 2023, 18 (6.7%) of whom accepted more than $5,000.

### Comparisons of 2014 and 2023 policies

When comparing the 2023 scores to their scores from 2014 based on the existing 14 domains, 5 schools (16.7%) maintained the same overall score, 11 (36.7%) had a greater overall score, and 14 (46.7%) had a lower overall score. The mean score for schools did not change significantly between 2014 and 2023 (32.7 versus 31.7; $p=0.14$). After omitting domain 7 (ghostwriting) which had a change in its grading criteria; a similar change was observed (29.9 versus 29.1; $p=0.17$). Of the 14 domains shared by both the 2014 and 2023 Scorecards, 9 domains (64.3%) had lower average scores

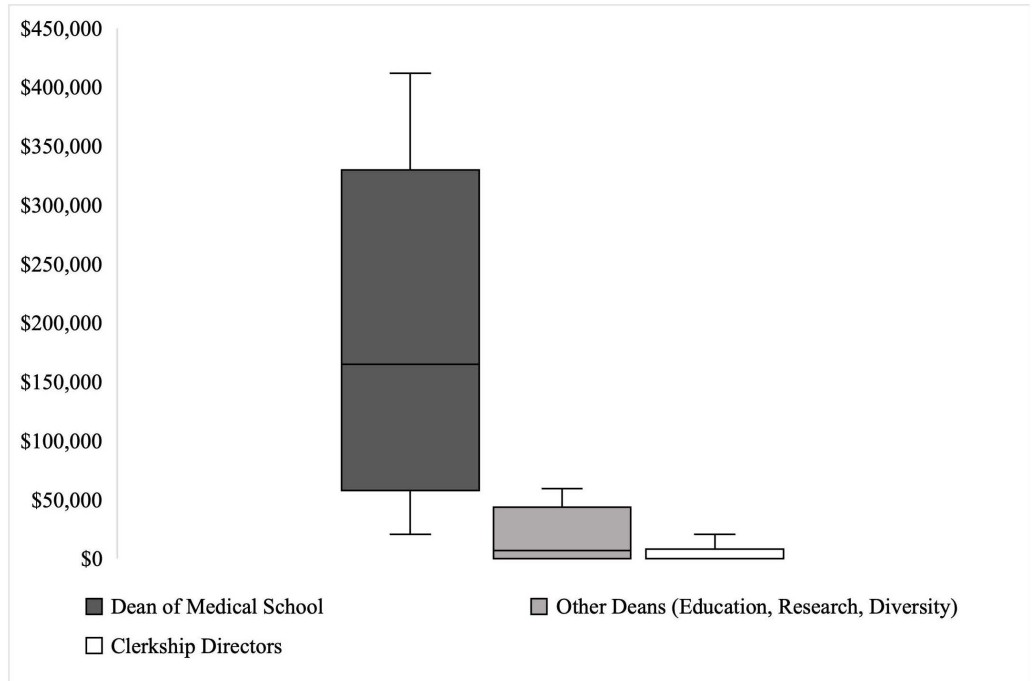

**Fig 3. Median and Interquartile Range for Medical School Leadership Receiving Industry Payments, 2023.** Box plots for medical school leadership receiving industry payments across all 30 medical schools, divided by dean of medical school (dark gray), other deans (light gray), and clerkship directors (white). A detailed breakdown of the "Other Deans" and "Clerkship Directors" categories is available in the supplemental material.

in 2023, while 5 (35.7%) had score increases (**Fig 4**). Of these, the only statistically significant change was a decrease in mean score for domain 5, attendance at industry-sponsored promotional events (2.1 versus 1.8; $p = 0.02$).

The number of schools achieving model policies reduced in multiple domains, including COI curricula (9 schools in 2014–1 school in 2023), consulting relationships (7 schools to 1 school), and attendance at industry-sponsored promotional events (8 schools to 2 schools) (**Fig 4**). Most policies did include safeguards to prevent industry influence and/or overcommitments outside of school responsibilities, but did not explicitly ban such activities, which was the grading requirement for a score of 3. Most schools also had vague or unclear COI curricula for medical students: there was either

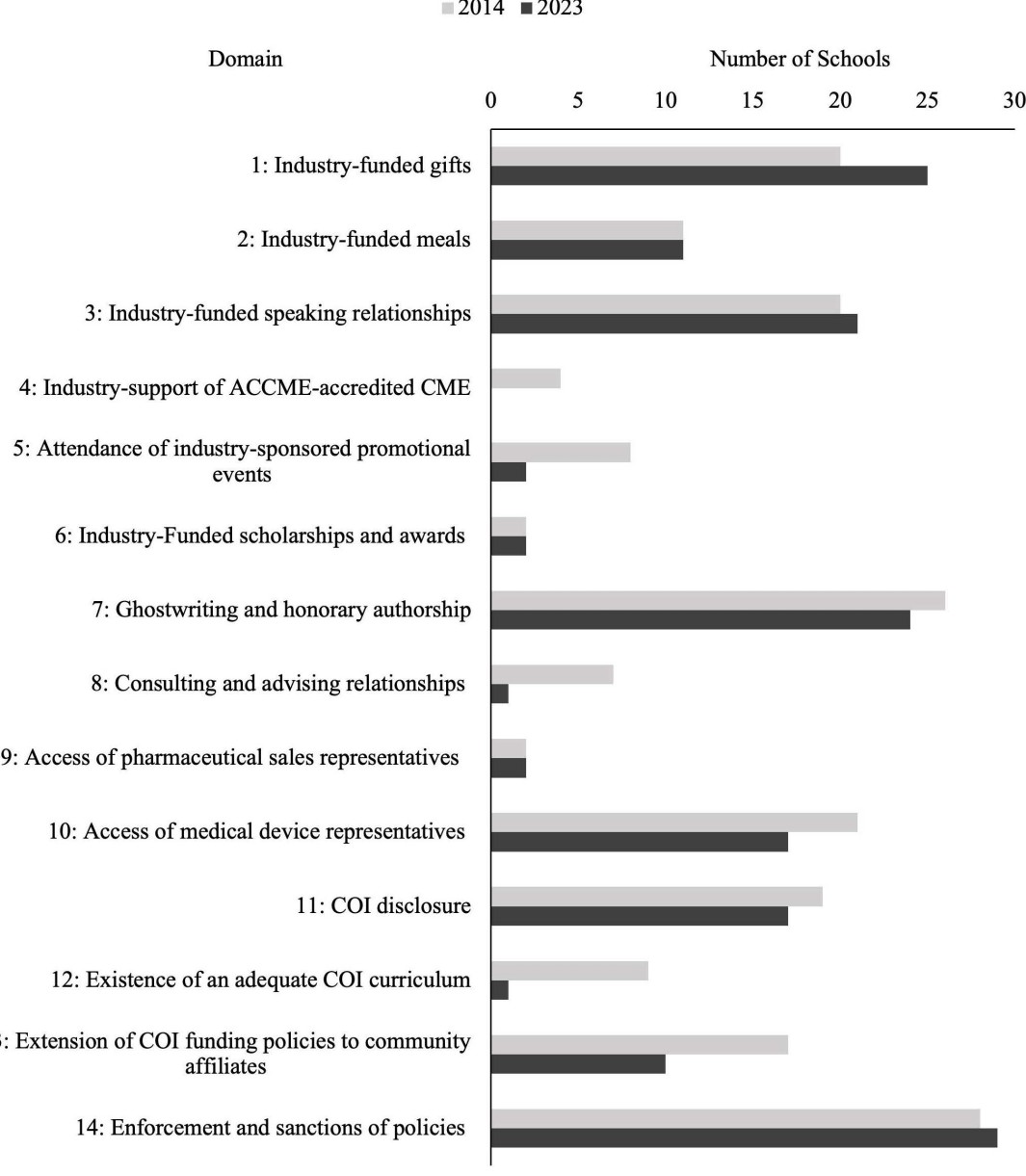

**Fig 4. Changes in Frequency of Model Conflict of Interest (COI) Policies in 2014 and 2023.** A comparison of the 30 schools' conflict of interest policies to model policies across the 14 consistent domains evaluated by this study in 2014 (gray) and 2023 (black).

brief mention of a required COI training, identification of COI courses without a detailed curriculum, or no curriculum or training mentioned at all (Fig 5).

## Discussion

The 2023 AMSA Scorecard identifies inadequate COI policies at top medical schools with limited change in the past decade. While schools scored well in several domains, such as the prohibition of industry gifts and ghostwriting, schools generally had inadequate policies. When comparing the 14 consistent domains in both the 2014 and 2023 Scorecards, many schools saw their average score drop. Schools also performed poorly on domain 15 (policies around industry payments to leadership), further lowering Scorecard grades this year. These results highlight the presence of financial ties between industry and medical schools and underline the overall need for stronger COI policies and enforcement.

This study also evaluated the financial interactions between industry and medical school leadership, offering deeper insights into indirect influences on medical education and research. Furthermore, while the majority of faculty overall did not receive industry funding, the uniformly lower scores in this new domain underscore the pervasive nature of potential conflicts and the need for policy enforcement. While the declining scores could be a by-product of efforts to collaborate with industry to produce new, innovative treatments, the consistent grading rubric over both the 2014 and 2023 iterations suggest continued risks of COI within medical education. Innovations that do arise from industry partnerships will be trusted more by patients and physicians if medical institutions have strong policies in place that address COI.

There are numerous potential causes for this year's Scorecard results. Perhaps the most plausible cause is direct payments to faculty—every medical school included in the survey had leadership receiving payments from industry. These findings are consistent with prior literature discussing receipt of industry funding by physicians at academic institutions. [25–28] These payments may support increased collaboration on clinical trials and development, help expand continuing medical education (CME), or fund labor or material costs of clinical research. Crucially, prior literature has repeatedly highlighted the ability of industry funding to bias this research, especially when payments are made directly to faculty rather

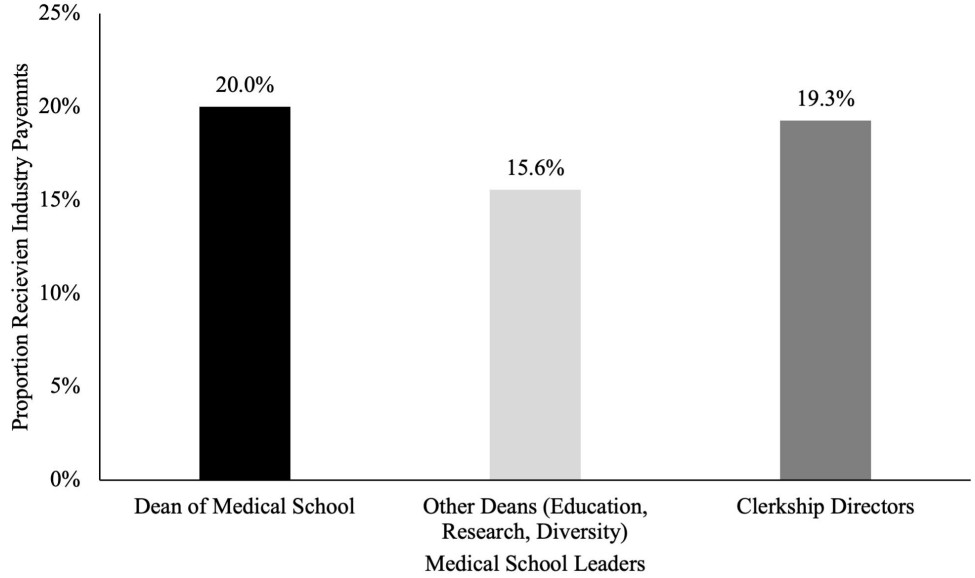

**Fig 5. Percentage of Medical School Leadership Receiving Industry Payments, 2023.** The percentage of total evaluated individuals across all 30 medical schools who received industry funding, divided by the dean of medical school (black), other deans (light grey), and clerkship directors (dark grey).

than the institution as a whole.[29] Given proposed federal funding cuts to critical research institutions (e.g., the National Institutes of Health), faculty and schools' dependence on industry funding for research programs may only worsen. These ties among leadership may also go beyond finances. While not explored in this study, faculty may bring a network of personnel ties to industry, further integrating medical education with industry and influencing the development of curriculum and COI policies. Transparency is also crucial: other studies have found that medical school leadership continue to accept payments and serve on industry boards, often without disclosing outside income.[28,30] Given this potential for these non-disclosures to impact patients in the long-term, medical schools should be incentivized to improve their COI policies. Federal regulations to encourage transparency (such as the creation of Open Payments) and enforce consequences for nondisclosures are crucial as well.

To address these potential conflicts, it is imperative for medical schools to adopt and enforce regulations that align with AAMC/NAM recommendations.[15,16] These should be aimed at recording and evaluating the nature of any direct payments faculty members receive from industry. Of note, this analysis focuses on written policies and does not consider the reality of industry payments on campus. Many surveys in the past decade have noted continued industry payments in practice, particularly for meals and lodging, despite the findings of this current study.[4,31,32] Ideally, universities would enact model policies that align with AAMC/NAM recommendations. However, we acknowledge that effective policies often involve gradual changes that partner with universities to protect the interests of both patients and providers.

Many academic medical centers have sought to strengthen COI policies. The University of Buffalo implemented a policy that eliminated industry-supplied meals and gifts, while integrating industry-sponsored research with academic oversight.[33] This policy also includes educating faculty and residents about the influence of industry marketing on clinical decision-making and requires disclosure of financial interests to patients, providing one example of how universities can make progress towards model policies.[33] Another similar initiative at Partners HealthCare (an academic affiliate of Harvard Medical School) creates an Education Review Board (ERB) which oversees and manages industry relationships. [34] Throughout the first few years of this program, ERB did not cause large reductions in the number of industry grants or the amount of funding, providing an additional example of a program universities can implement AAMC/NAM recommendations while preserving funding and innovation.[34]

Despite over a decade of clear guidance from academic medical organizations encouraging limitations on industry interaction with medical trainees and faculty, recent evidence on these relationships and patient outcomes is limited, with critics suggesting these dialogues may stifle the pharmaceutical industry's contributions to innovation.[35–38] Often these concerns conflate academic-industry research relationships (e.g., industry support for a clinical trial) with industry marketing relationships. However, receipt of industry marketing payments is associated with increased medical and prescription drug costs, which will ultimately fall back to patients via higher premiums and copayments.[39–41] For example, industry payments have been shown to increase prescription rates of non-recommended cancer therapies for prostate cancer, a finding that may have implications on patient outcomes.[40] Thus, while physician-industry relationships and their implications pertaining to patient care continue to be debated, current literature does indicate the potential for harm to both the healthcare system (e.g., higher costs) and patients (e.g., differences in medication management) specifically for marketing-based interactions. Future research is needed to delineate the ideal balance that allows industry-academia research advancements to continue without compromising the integrity of patient care.

Findings from other countries paint a similar picture: recent literature has highlighted that medical schools in Australia, Canada, and select European countries have adopted some form of COI policies, but that these policies are often limited in scope, lack standardized reporting requirements, and very widely in enforcement.[42–46] Despite national guidelines in certain countries, institutional implementation and oversight remains incomplete. Curricula around COI, including ensuring medical students are educated on how to handle ties with industry, is often lacking, along with inconsistent policies around acceptance of gifts, access to sales representatives, and other shared domains. These parallels suggest that the

challenges identified in American medical schools, heterogeneity and partial adherence to best practices, reflect broader international struggles to establish comprehensive and enforceable COI frameworks.

Evidence suggests that stronger institutional restrictions can meaningfully reduce student-industry interactions. Students at schools with more restrictive COI policies reported fewer interactions with industry representatives, demonstrated more critical attitudes towards pharmaceutical marketing, and as resident physicians, prescribed new medications at lower rates.[47] More investigation is required on the changes in prescribing habits, but it is clear that well-designed COI policies can indeed reduce exposure to potential bias early in training.

This study has several limitations. Firstly, the selection of 30 medical schools was reliant on U.S. News Rankings; as of 2022, many of the included medical schools withdrew from the ranking process due to concerns about the methodology.[48–50] These concerns included overemphasizing the role of standardized test scores and other numeric grades. [48] Because this analysis focused on the thirty top-ranked American medical schools, the findings may not represent broad policy strength across medical schools. Higher-ranked schools may be subject to greater public scrutiny, potentially incentivizing stronger COI policies. Conversely, it also possible that top-ranked programs have more extensive research partnerships and funding, giving them greater exposure to industry. Future studies should expand this analysis to a broader set of schools to assess whether patterns identified here are consistent across the full landscape of U.S. medical education.

Furthermore, the presence of COI policies does not mean that institutions are adhering to them. While schools had strong enforcement mechanisms, it is hard to independently verify this enforcement. A landmark study published in 2013 found that medical students interacted with the pharmaceutical industry extensively, through meals, gifts, and direct communication by representatives.[51] This exposure to industry rose throughout the course of medical school and often led to improved views of industry ties. More recent work has found that medical students remain concerned about and underprepared in dealing with potential COIs from industry exposure.[52] Crucially, this industry exposure, and the impact it can have on student perception of industry, is present even at schools with stringent COI policies. To ensure COI policies truly impact behavior, medical schools should consider mandatory COI curricula for faculty and students, routine audits of industry payments, and inclusion of COI compliance in faculty evaluations. The response rate of schools to this survey study was also low, and the authors relied on publicly posted policies which may not reflect the most recent policies enacted by schools. However, these do represent the policies accessible by both pharmaceutical companies and by prospective medical students considering matriculation. Additionally, this study did not include an analysis of institutional characteristics, such as public or private status or degree of federal funding dependence, which may influence development of COI policies. Further research could examine these factors to better understand potential drivers of policy variation. As a survey of COI policies, this study did not consider the first-hand experiences of medical students and residents, and future research may explore the implementation and impact of COI policies at medical schools through surveys of students and faculty.[15,52,53] Finally, while this study focuses on policies implemented by medical schools, industry actors also maintain codes of conduct that govern their interactions with academic institutions. In the United States, the Pharmaceutical Research and Manufacturers of America (PhRMA) Code on Interactions with Health Care Professionals provides guidance limiting meals, outlining appropriate consulting relationships, and educational partnerships.[54] However, adherence to these self-regulatory codes is largely voluntary and varies across companies. Evaluating industry adherence to these standards, and improvement of the standards themselves, remains an important area for future research.

## Conclusion

The 2023 AMSA Scorecard identified several opportunities for improved policies within all medical schools evaluated. We believe that these findings will encourage schools to modify their COI policies, safeguarding the integrity of medical education. A bias-free education will help shape physicians that are independent, evidence-based prescribers for patients, ultimately improving patient care. The Scorecard's impact will be monitored on multiple avenues: we will analyze public

changes that schools make in response to publication and collaborate with AMSA chapters at individual schools to stay up to date on potential policy changes.

## Supporting information

**S1 Appendix. Email and Corresponding Survey Questions Sent to Top 30 Medical Schools.**
(DOCX)

**S1 Table. List of Medical Schools Included in 2023.**
(DOCX)

**S2 Table. Source of Policies Analyzed for Schools.**
(DOCX)

**S2 Appendix. Medical School Conflict of Interest Policy Domains and Model Policy Definitions.**
(DOCX)

**S3 Appendix. 2023 PharmFree Scoring Methodology.**
(DOCX)

**S3 Table. Mean Total Scores by Policy Submission Status, 2023.**
(DOCX)

**S4 Figure. A Closer Look at Medical School Deans/Clerkship Directors.**
(DOCX)

## Acknowledgments

The abstract was presented as both a poster and an oral presentation at the 2025 Accreditation Council for Graduate Medical Education (ACGME) Conference on February 20th, 2025 in Nashville, TN.

## Author contributions

**Conceptualization:** Shamik Bhat, Devika A. Shenoy, Brianna Wang, Timothy S. Anderson.

**Data curation:** Shamik Bhat, Devika A. Shenoy, Magda Wojtara, Alissa Kainrath, Jantzen Faulkner.

**Formal analysis:** Shamik Bhat, Devika A. Shenoy, Alissa Kainrath, Jantzen Faulkner, Linnea Wilson.

**Funding acquisition:** Shamik Bhat, Oak Sonfist.

**Investigation:** Shamik Bhat, Devika A. Shenoy.

**Methodology:** Shamik Bhat, Devika A. Shenoy, Magda Wojtara.

**Project administration:** Shamik Bhat, Devika A. Shenoy, Oak Sonfist, Timothy S. Anderson.

**Supervision:** Shamik Bhat, Oak Sonfist, Timothy S. Anderson.

**Writing – original draft:** Shamik Bhat, Devika A. Shenoy, Magda Wojtara, Alissa Kainrath.

**Writing – review & editing:** Shamik Bhat, Devika A. Shenoy, Timothy S. Anderson.

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
