## [Decision Letter · Decision Letter 0]

1 Oct 2025

Dear Dr. Bhat,

We look forward to receiving your revised manuscript.

Kind regards,

Avanti Dey, PhD

Staff Editor

PLOS ONE

Journal Requirements:

Additional Editor Comments (if provided):

Reviewers' comments:

Reviewer's Responses to Questions

**Comments to the Author**

1. Is the manuscript technically sound, and do the data support the conclusions?

Reviewer #1: Yes

Reviewer #2: Yes

2. Has the statistical analysis been performed appropriately and rigorously?

Reviewer #1: N/A

Reviewer #2: Yes

3. Have the authors made all data underlying the findings in their manuscript fully available?

Reviewer #1: Yes

Reviewer #2: Yes

4. Is the manuscript presented in an intelligible fashion and written in standard English?

Reviewer #1: Yes

Reviewer #2: Yes

Reviewer #1: This renewal of the AMSA scorecard about COI policies at US medical schools is a welcome update that allows readers to see whether there has been any significant change in the relationship between medical schools and industry.

I would encourage the authors to expand their discussion section to include two additional topics. How do policies at US medical schools compare with policies at schools in other countries. There is now published material about policies in Australia, Belgium, Canada, France and Scandinavia. Second, the authors should discuss the effect of policies at medical schools that restrict interactions between students and industry – see for example King et al. BMJ 2013;346:f264.

Line 41: The authors should make it clear if they are talking about about education of medical students, education of trainees, continuing medical education/professional development or all three?

Lines 72-75: The authors should make it clear that they are talking about the situation in the United States as the situation in other countries may be different.

Line 88: Reference 13 by Lexchin is over 30 years old. I would suggest that the authors use the following reference instead: Spurling GK, Mansfield PR, Montgomery BD, Lexchin J, Doust J, Othman N, Vitry AI. Information from pharmaceutical companies and the quality, quantity, and cost of physicians’ prescribing: a systematic review. PLoS Medicine 2010;7:e1000352.

Line 94: AMSA is now accepting funding from drug companies - https://www.statnews.com/pharmalot/2024/06/05/conflicts-pharmaceuticals-universities/#:~:text=In%20a%20little%2Dnoticed%20move,value%20without%20direction%20or%20conditions.”&text=The%20board%20explained%20in%20April,“integrity%20of%20our%20work.”

Line 97: When the authors say that "much has changed in medicine" what specifically are they referring to? Many of these changes, including ones that are overall detrimental to healthcare, e.g., increases in equity-ownership of health care facilities, may have little to do with physician-industry relationships.

Line 108: What criteria are used for the rankings done by US News and World Report?

Line 174: Do the authors mean schools that submitted policies directly to them, not to AMSA?

Reviewer #2: General comments

Thank you very much for the opportunity to review this interesting article. While descriptive, the paper contributes to the literature on conflicts of interest in healthcare by assessing the COI policies enacted by a selected number of medical schools.

I have some general comments that the authors might want to take on board:

(1) While I understand why you selected the universities included in the study, and you provide the rationale for doing so, it would be useful to discuss how these might differ from others not included in the sample. For example, non–top-ranked medical schools might be less scrutinized and have less stringent policies, or, alternatively, the might be less prone to capture by industry.

(2) I think you should expand the discussion section where you acknowledge that there are policies and enforcement mechanisms. You briefly touch upon that, but it would be useful to understand better whether these policies are actually reflected in faculty and student behaviors, and what reforms should be in place to ensure that relationships with industry are managed appropriately de facto (and not only on paper).

(3) I would also be interested to know whether industry has similar policies on how they themselves interact with medical univesities.

Introduction

page 4, line 101: Please clarify whether the physicians from whom you extract information teach or have administrative roles.

Methods

page 5, line 124: Sometimes you refer to the previous scorecard version as 2014 and sometimes as 2016. I understand these dates refer to data collection and publication, but I would be consistent and use just one.

page 5, line 136: Please elaborate on why ghostwriting is problematic, as it is not immediately clear.

page 6, lines 152–155: Please clarify that these values refer to 2023. Also, are these declared in 2023 or received in 2023? This is an important clarification, although unlikely to affect your results.

page 6, lines 155–158: Please explain whether the research assistants also graded policies. It is unclear what “assessed each policy” currently means.

Results

page 8, line 191: Have you thought about exploring other variables in the analysis? Types of school or dependency on federal funding could be relatively easy variables to control for, which could add more nuance to the analysis.

page 8, line 199: You can say “no schools” instead of “few,” and “just one” for faculty having consulting relationships.

page 9, line 212: Please add that these values refer to 2023.

page 10, lines 236–238: It is a bit unclear why you decided to report proportions this way. Also, why is the total different across categories?

Discussion

page 11, line 255: I am not clear how you can compare the average score between the two iterations if you do not remove the additional category you included. Or is this category already excluded here?

page 11, line 269: I think your findings on payments to faculty are very interesting. It would be useful to understand what you think the origin of these is—is this some kind of compensation mechanism to offset government funding cuts? Does this risk getting worse in the current context of a reduced NIH budget?

**Do you want your identity to be public for this peer review?** For information about this choice, including consent withdrawal, please see our Privacy Policy

Reviewer #1: **Yes:** Joel Lexchin

Reviewer #2: No

---

## [Author Response · Author response to Decision Letter 1]

3 Nov 2025

To the Editors of PLOS ONE,

REVISION REQUIRED PONE-D-25-32688

Please see our rebuttal letter below that includes responses to every point raised in your comments. We have sought to address your concerns to the best of our abilities—please let us know if there are additional steps we can take. Your comments are bolded, with our responses below.

Reviewer 1:

1. I would encourage the authors to expand their discussion section to include two additional topics. How do policies at US medical schools compare with policies at schools in other countries. There is now published material about policies in Australia, Belgium, Canada, France and Scandinavia. Second, the authors should discuss the effect of policies at medical schools that restrict interactions between students and industry – see for example King et al. BMJ 2013;346:f264.

We appreciate this suggestion. We have added two paragraphs to the text addressing these topics. Specifically, we now mention the shared challenges in COI policies in the US, Australia, and European countries. We also reference King et al. to highlight the effect of stringent policies on student views and behaviors towards the pharmaceutical industry (discussion, 339-354).

2. Line 41: The authors should make it clear if they are talking about about education of medical students, education of trainees, continuing medical education/professional development or all three?

Our domains of analysis include all three components; the text has been clarified (abstract, line 42).

3. Lines 72-75: The authors should make it clear that they are talking about the situation in the United States as the situation in other countries may be different.

Thank you—clarification has been added (introduction, lines 74, 75).

4. Line 88: Reference 13 by Lexchin is over 30 years old. I would suggest that the authors use the following reference instead: Spurling GK, Mansfield PR, Montgomery BD, Lexchin J, Doust J, Othman N, Vitry AI. Information from pharmaceutical companies and the quality, quantity, and cost of physicians’ prescribing: a systematic review. PLoS Medicine 2010;7:e1000352.

Thank you—this reference has been updated.

5. Line 94: AMSA is now accepting funding from drug companies - https://www.statnews.com/pharmalot/2024/06/05/conflicts-pharmaceuticals-universities/#:~:text=In%20a%20little%2Dnoticed%20move,value%20without%20direction%20or%20conditions.”&text=The%20board%20explained%20in%20April,“integrity%20of%20our%20work.”

This was a disappointing change in policy. This study predates that change, however, and is not influenced or connected to industry funding in any way. This is an important concern, however, and provides further context to the changing climate of COI policies.

6. Line 97: When the authors say that "much has changed in medicine" what specifically are they referring to? Many of these changes, including ones that are overall detrimental to healthcare, e.g., increases in equity-ownership of health care facilities, may have little to do with physician-industry relationships.

Details on additional industry-academic ties have been added to the text (introduction, lines 100-104).

7. Line 108: What criteria are used for the rankings done by US News and World Report?

Details on the 2022 US News and World Report’s research rankings are now included in the text (introduction, 116-117).

8. Line 174: Do the authors mean schools that submitted policies directly to them, not to AMSA?

Schools could choose to submit policies to the team via the AMSA PharmFree website; these were available for the team to analyze. This is clarified in the text (statistical analysis, line 188).

Reviewer 2:

1. I have some general comments that the authors might want to take on board:

While I understand why you selected the universities included in the study, and you provide the rationale for doing so, it would be useful to discuss how these might differ from others not included in the sample. For example, non–top-ranked medical schools might be less scrutinized and have less stringent policies, or, alternatively, the might be less prone to capture by industry.

Thank you for this suggestion. We have added text to the discussion section (lines 358-365) to reflect this possibility, acknowledging that our analysis only included leading medical schools and that a broader analysis could reach different conclusions. Past AMSA Scorecards have been broader, and we have suggested expanding future work in this direction to assess the generalizability of our findings.

2. I think you should expand the discussion section where you acknowledge that there are policies and enforcement mechanisms. You briefly touch upon that, but it would be useful to understand better whether these policies are actually reflected in faculty and student behaviors, and what reforms should be in place to ensure that relationships with industry are managed appropriately de facto (and not only on paper).

Thank you for this suggestion. We have expanded the discussion section (lines 368-375) to include consideration of the impact of COI policies, policies’ impact on student behavior and perception, and how schools can better enforce the policies they implement.

3. I would also be interested to know whether industry has similar policies on how they themselves interact with medical univesities.

Thank you for this suggestion. We have added text to the discussion section (lines 385-392) briefly summarizing industry guidance on interactions with health care professionals. We note that the guidelines are voluntary and adherence to them is a key area of future research.

4. Introduction

page 4, line 101: Please clarify whether the physicians from whom you extract information teach or have administrative roles.

Further detail regarding physicians analyzed is included (introduction, lines 108-109), and their analysis regardless of teaching status is later clarified (methods, line 160).

5. Methods

page 5, line 124: Sometimes you refer to the previous scorecard version as 2014 and sometimes as 2016. I understand these dates refer to data collection and publication, but I would be consistent and use just one.

Thank you—this has been updated throughout the text.

6. page 5, line 136: Please elaborate on why ghostwriting is problematic, as it is not immediately clear.

Thank you for this suggestion—ghostwriting concerns are now elaborated in the text (methods, line 145-147).

7. page 6, lines 152–155: Please clarify that these values refer to 2023. Also, are these declared in 2023 or received in 2023? This is an important clarification, although unlikely to affect your results.

Received in 2023; this clarification is added in the text (methods, lines 165-166).

8. page 6, lines 155–158: Please explain whether the research assistants also graded policies. It is unclear what “assessed each policy” currently means.

They assessed policies according to AMSA model policies and then graded each policy; this clarification is added to the text (methods, lines 169-170).

9. Results

page 8, line 191: Have you thought about exploring other variables in the analysis? Types of school or dependency on federal funding could be relatively easy variables to control for, which could add more nuance to the analysis.

Thank you for this feedback. While these factors may indeed influence conflict-of-interest policy strength, our study was primarily designed as a descriptive comparison of publicly available policies and did not collect covariates, such as type of school or funding sources. Future work should certainly incorporate these characteristics to further analysis COI policy variation, and we have included this in the text to reflect this (discussion, lines 379-382).

10. page 8, line 199: You can say “no schools” instead of “few,” and “just one” for faculty having consulting relationships.

Thank you for highlighting this—text has been corrected (results, lines 214-217).

11. page 9, line 212: Please add that these values refer to 2023.

This has been clarified throughout the paragraph (results, lines 227-231).

12. page 10, lines 236–238: It is a bit unclear why you decided to report proportions this way. Also, why is the total different across categories?

This has been clarified; we are reporting the number of schools with model policies in each domain and the drop in number from 2014 to 2023 (results, lines 252-253).

13. Discussion

page 11, line 255: I am not clear how you can compare the average score between the two iterations if you do not remove the additional category you included. Or is this category already excluded here?

Thank you for this observation. Our comparisons of 2014 and 2023 average scores exclude the additional domain (domain 15). These results hold true just comparing the 14 consistent domains, and the text has been clarified on this point (discussion, lines 269-272).

14. page 11, line 269: I think your findings on payments to faculty are very interesting. It would be useful to understand what you think the origin of these is—is this some kind of compensation mechanism to offset government funding cuts? Does this risk getting worse in the current context of a reduced NIH budget?

Thank you for this insight. We acknowledge additional investigation into the origin of industry payments to physicians would be enlightening and that federal funding cuts could certainly worsen schools’ dependence on industry. The relevant paragraph has been updated (discussion, lines 288-294).

---

## [Decision Letter · Decision Letter 1]

25 Jan 2026

Dear Dr. Bhat,

Thank you for submitting your manuscript to PLOS ONE. After careful consideration, we feel that it has merit but does not fully meet PLOS ONE’s publication criteria as it currently stands. Therefore, we invite you to submit a revised version of the manuscript that addresses the points raised during the review process.

Dear Authors kindly address all the reviewers suggestions.

We look forward to receiving your revised manuscript.

Kind regards,

Ramya Iyadurai

Academic Editor

PLOS One

Journal Requirements:

Reviewers' comments:

Reviewer's Responses to Questions

**Comments to the Author**

Reviewer #1: All comments have been addressed

Reviewer #3: All comments have been addressed

2. Is the manuscript technically sound, and do the data support the conclusions?

Reviewer #1: Yes

Reviewer #3: Partly

3. Has the statistical analysis been performed appropriately and rigorously?

Reviewer #1: Yes

Reviewer #3: Yes

4. Have the authors made all data underlying the findings in their manuscript fully available?

Reviewer #1: Yes

Reviewer #3: Yes

5. Is the manuscript presented in an intelligible fashion and written in standard English?

Reviewer #1: Yes

Reviewer #3: Yes

Reviewer #1: I appreciate the changes that the authors have made. There are still some additional issues that need to be addressed.

1. Page 2, line 66: The mention of "national guidance" needs to be explained as it is the first time in the Abstract that the term has been mentioned.

2. Reference 9 for the percent of CME money coming from industry is for 2017. There are newer data, for 2024, which puts industry revenue (commercial support + advertising revenue) at about 42% of total income. The correct URL is https://accme.org/wp-content/uploads/2025/06/2024accmeannualdatareport1077_20250630-1.pdf

3. Page 4, lines 99-100: The authors say that the last scorecard was published in 2014 but on page 5, line 138 they say it was published in 2016.

4. Page 6, line 143: There should be a reference about the influence of ghostwriting. I'd suggest Lacasse JR, Leo J (2010) Ghostwriting at Elite Academic Medical Centers in the United States. PLoS Med 7(2): e1000230.

5. Page 7, line 167-168: What was the rationale for how the categories of the amount of industry payments to faculty was chosen?

6. Page 14, line 342: Reference 42 refers to COI of Canadian medical school deans not COI policies of Canadian medical schools.

7. Page 16, line 370: Reference 49 is over a decade old. The authors should either justify its current relevance or try to replace it with a more current reference.

Minor points

1. Page 4, line 107: It should be "physicians".

Reviewer #3: I believe this is an excellent contribution to the field. You have highlighted the importance of COI policies through a rigorous and well-reported descriptive study. Please view this submission not as a conclusion, but as a foundation; I look forward to seeing the future work mentioned in your manuscript.

**Do you want your identity to be public for this peer review?** For information about this choice, including consent withdrawal, please see our Privacy Policy

Reviewer #1: **Yes:** Joel Lexchin

Reviewer #3: **Yes:** Mahmoud Bassiony

---

## [Author Response · Author response to Decision Letter 2]

5 Feb 2026

February 5, 2026

To the Editors of PLOS ONE,

REVISION REQUIRED PONE-D-25-32688

We thank all reviewers and editors for their feedback, which we believe has helped improve our manuscript. Please see our response letter below that includes a description of how we addressed reviewer feedback/comments. We remain available for additional changes as needed and look forward to the prospect of contributing to PLoS One. Please note that line numbers below refer to the “TRACKED” version of the manuscript.

REVIEWER 1

Reviewer 1, Comment #1: The mention of "national guidance" needs to be explained as it is the first time in the Abstract that the term has been mentioned.

• Author Response: Thank you for highlighting this area for improvement. Guidelines have been elaborated on briefly earlier in the abstract, and the term itself has been removed from the abstract’s conclusion.

o Text Change 1: “Using previously validated methodologies, the authors collected policies by survey and review of public websites, and assessed their quality across 15 domains informed by guidelines published by leading national organizations and previous PharmFree Scorecards.” (Abstract, lines 50-51)

o Text Change 2: “Medical school COI policies remain less stringent than consensus recommendations” (Abstract, lines 67-68)

Reviewer 1, Comment #2 : Reference 9 for the percent of CME money coming from industry is for 2017. There are newer data, for 2024, which puts industry revenue (commercial support + advertising revenue) at about 42% of total income. The correct URL is https://accme.org/wp-content/uploads/2025/06/2024accmeannualdatareport1077_20250630-1.pdf

• Author Response: Thank you for linking updated data. We have updated the references to reflect the most recent findings; it is now reference 14.

Reviewer 1, Comment #3: The authors say that the last scorecard was published in 2014 but on page 5, line 138 they say it was published in 2016.

• Author Response: Thank you for raising this point for further clarification. The last Scorecard was completed in 2014, but publication stretched on to 2016. This has been clarified in the text by removing the publication year, as it is an unnecessary detail that confuses the reader.

o Text Change: “The most recent prior iteration of the AMSA Scorecard analyzed COI policies enacted in or before 2014.” (Methods, line 142)

Reviewer 1, Comment #4: There should be a reference about the influence of ghostwriting. I'd suggest Lacasse JR, Leo J (2010) Ghostwriting at Elite Academic Medical Centers in the United States. PLoS Med 7(2): e1000230.

• Author Response: Thank you for this suggestion. This reference has been added (reference 22) and is now cited in the relevant section (Methods, line 152).

Reviewer 1, Comment #5: What was the rationale for how the categories of the amount of industry payments to faculty was chosen?

• Author Response: We appreciate this thoughtful question. We have now provided clarification: administrative rules published by the Department of Health and Human Services (HHS) have consistently considered yearly contributions of $5,000 for researchers and physicians as “significant financial interests.” We thus chose $5,000 to remain consistent with federal definitions. Citations to federal guidance are now added to the relevant section

o Text Changes: “Federal rules published by the Department of Health and Human Services generally define yearly contributions of $5,000 as reaching the threshold of “significant financial interests” for researchers and clinicians. We adopt the $5,000 yearly benchmark to remain consistent with this guidance.[23, 24]” (Methods, lines 173-176).

Reviewer 1, Comment #6: Reference 42 refers to COI of Canadian medical school deans not COI policies of Canadian medical schools.

• Author Response: Thank you for highlighting this. Given that reference 43 is a similar analysis of COI policies at Canadian medical schools (similar to PharmFree) and thus more relevant, we have removed reference 42.

Reviewer 1, Comment #7: Reference 49 is over a decade old. The authors should either justify its current relevance or try to replace it with a more current reference.

• Author Response: Thank you for raising this point. While the reference is old, it is a landmark study documenting extensive industry-student interactions, and there is limited work documenting this interaction more recently. More recent work has continued to find that potential conflicts of interest are of concern to medical students, and we have added this additional context to the manuscript. We have also mentioned that the study was published in 2013.

o Text Changes: “A landmark study published in 2013 found that medical students interacted with the pharmaceutical industry extensively, through meals, gifts, and direct communication by representatives.[51] This exposure to industry rose throughout the course of medical school and often led to improved views of industry ties. More recent work has found that medical students remain concerned about and underprepared in dealing with potential COIs from industry exposure.[52]” (Discussion, lines 376-381)

Reviewer 1, Comment #8: It should be "physicians".

• Author Response: Thank you for noting this error. This has now been fixed (Introduction, line 106).

REVIEWER 3

Reviewer 3, Comment #1: I believe this is an excellent contribution to the field. You have highlighted the importance of COI policies through a rigorous and well-reported descriptive study. Please view this submission not as a conclusion, but as a foundation; I look forward to seeing the future work mentioned in your manuscript.

• Author Response: Thank you for this positive feedback. We agree and believe that this work is the first step in reestablishing regular COI policy analysis in medical education. We hope and intend to continue and expand this analysis to all medical schools in the United States, and later include metrics of implementation (e.g., student surveys, etc.).

---

## [Editor Report · Decision Letter 2]

16 Feb 2026

Changes in U.S. Medical School Conflict of Interest Policies from 2014 to 2023

PONE-D-25-32688R2

Dear Dr. Bhat,

We’re pleased to inform you that your manuscript has been judged scientifically suitable for publication and will be formally accepted for publication once it meets all outstanding technical requirements.

Kind regards,

Ramya Iyadurai

Academic Editor

PLOS One

Additional Editor Comments (optional):

None
---

## [Editor Report · Acceptance letter]

PONE-D-25-32688R2

PLOS One

Dear Dr. Bhat,

I'm pleased to inform you that your manuscript has been deemed suitable for publication in PLOS One. Congratulations! Your manuscript is now being handed over to our production team.

Kind regards,

on behalf of

Dr. Ramya Iyadurai

Academic Editor

PLOS One